# Engineered EV-Mimetic Nanoparticles as Therapeutic Delivery Vehicles for High-Grade Serous Ovarian Cancer

**DOI:** 10.3390/cancers13123075

**Published:** 2021-06-20

**Authors:** Amal A. Al-Dossary, Essam A. Tawfik, Adaugo C. Isichei, Xin Sun, Jiahe Li, Abdullah A. Alshehri, Munther Alomari, Fahad A. Almughem, Ahmad M. Aldossary, Hussein Sabit, Abdulaziz M. Almalik

**Affiliations:** 1Department of Basic Sciences, Deanship of Preparatory Year and Supporting Studies, Imam Abdulrahman Bin Faisal University, P.O. Box 1982, Dammam 34212, Saudi Arabia; aisichei@iau.edu.sa; 2National Center for Pharmaceutical Technology, Life Science and Environment Research Institute, King Abdulaziz City for Science and Technology (KACST), P.O. Box 6086, Riyadh 11442, Saudi Arabia; etawfik@kacst.edu.sa (E.A.T.); abdualshehri@kacst.edu.sa (A.A.A.); falmughem@kacst.edu.sa (F.A.A.); aalmalik@kacst.edu.sa (A.M.A.); 3Department of Bioengineering, Northeastern University, Boston, MA 02115, USA; sun.xin1@northeastern.edu (X.S.); jiah.li@northeastern.edu (J.L.); 4Department of Stem Cell Biology, Institute for Research and Medical Consultations (IRMC), Imam Abdulrahman Bin Faisal University, P.O. Box 1982, Dammam 31441, Saudi Arabia; maomari@iau.edu.sa; 5National Center of Biotechnology, Life Science and Environment Research Institute, King Abdulaziz City for Science and Technology (KACST), P.O. Box 6086, Riyadh 11442, Saudi Arabia; aaldossary@kacst.edu.sa; 6Department of Genetics Research, Institute for Research and Medical Consultations (IRMC), Imam Abdulrahman Bin Faisal University, P.O. Box 1982, Dammam 31441, Saudi Arabia; hhsabit@iau.edu.sa

**Keywords:** high-grade serous ovarian cancer, peritoneal dissemination, EVs, engineered EV-mimetic nanoparticles, drug delivery, chemotherapy, gene therapy, nanotechnology, clinical translation

## Abstract

**Simple Summary:**

In this review, we begin with the role of natural extracellular vesicles (EVs) in high-grade serous ovarian cancer (HGSOC). Then, we narrow our focus on the advantages of using EV-mimetic nanoparticles as a delivery vehicle for RNAi therapy and other chemotherapeutics. Furthermore, we discuss the challenges of the clinical translation of engineering EV mimetic drug delivery systems and the promising directions of further development.

**Abstract:**

High-grade serous ovarian cancer (HGSOC) is the most lethal gynecological malignancy among women. Several obstacles impede the early diagnosis and effective treatment options for ovarian cancer (OC) patients, which most importantly include the development of platinum-drug-resistant strains. Currently, extensive efforts are being put into the development of strategies capable of effectively circumventing the physical and biological barriers present in the peritoneal cavity of metastatic OC patients, representing a late stage of gastrointestinal and gynecological cancer with an extremely poor prognosis. Naturally occurring extracellular vesicles (EVs) have been shown to play a pivotal role in progression of OC and are now being harnessed as a delivery vehicle for cancer chemotherapeutics. However, there are limitations to their clinical application due to current challenges in their preparation techniques. Intriguingly, there is a recent drive towards the use of engineered synthetic EVs for the delivery of chemotherapeutics and RNA interference therapy (RNAi), as they show the promise of overcoming the obstacles in the treatment of OC patients. This review discusses the therapeutic application of EVs in OC and elucidates the potential use of engineered EV-mimetic nanoparticles as a delivery vehicle for RNAi therapy and other chemotherapeutics, which would potentially improve clinical outcomes of OC patients.

## 1. Introduction

### 1.1. Clinical Significance of Ovarian Cancer

Ovarian epithelial cancer remains the most lethal gynecological malignancy among women worldwide [1,2]. In 2020, an estimated 207,252 OC patients died from this cancer worldwide and there were approximately 313,959 newly diagnosed cases [3]. The vast majority of the ovarian OC deaths are related to advanced clinical stages of high-grade serous ovarian cancer (HGSOC) [4]. HGSOC is type II epithelial OC and is characterized by high malignity, aggressive spread with extreme sensitivity and resistance to chemotherapy [5,6,7,8]. There are five molecular subtypes of HGSOC: mesenchymal, anti-mesenchymal, differentiated, proliferative, and immunoreactive [9,10,11]. In addition, Feng et al. [12] classified HGSOC based on hormone receptor expression. The expression level of hormone receptors: ER, PR, AR, FSHR, LHR, and GnRHR has been detected in HGSOC tissues as following 64.4%, 12.6%, 35.6%, 54.5%, 34.8%, and 88.3%, respectively. The poor prognosis and survival outcomes of OC patients have not been significantly altered in recent decades. When diagnosed at an early stage, the five-year survival rate is over 90 percent. However, the five-year survival rate of advance stage OC patients is approximately 20 percent [13,14,15,16]. Thus, the development of effective therapeutic approaches is of the utmost urgency. Ascetic fluid facilitates the entry of OC cells into the lymphatic and that makes it a favorable environment for wide peritoneal metastasis (PM) [17]. Moreover, HGSOC does not require the blood or lymph to metastasize, as it can aggressively spread by direct contact to the neighboring organs within the peritoneal cavity [8]. The mechanism underlying the peritoneal implantation of cancer cells is unknown. Notably, research on extracellular nanovesicles (EVs) has come under the spotlight as novel mediators of tumor metastasis that promote PM and also are potential therapeutics for OC [18]. In general, EVs are bilayer lipid vesicles that contain various bioactive molecules, such as microRNAs (miRNAs), messenger RNAs (mRNAs) and proteins [19,20,21,22,23], as shown in Figure 1, highlighting their potential clinical applications such as therapeutic targets, diagnostic biomarkers, and drug-delivery vehicles [24]. Subsequently, it has been shown that these EVs carry matrix metalloproteinase-1 (MMP1) mRNA, which is a key molecule that promotes PM [18]. Yoshimura and colleagues [25] demonstrated that metastatic OC secreted EVs carrying miR-99a-5p, which effectively prompt cell invasion and cancer progression by upregulation of fibronectin and vitronectin expression levels. Recently, Pinto et al. [26] showed that the use of EVs derived from mesenchymal stromal cells in combination with a fourth-generation photodynamic therapeutic (PDT) agent and an immune activator for the treatment of PM showed great promise in augmenting the antitumor immune response, reducing tumor proliferation, minimizing systemic toxicity of the PDT target with superior selectivity, and enhancing survival rate. Overall, these observations support the importance of EVs as a bio-inspired approach to treatment of PM.

### 1.2. What Are EVs?

EVs (mainly exosomes) have gained increasing attention due to their composition of highly specialized biological entities involved in long-distance intracellular communication during a multitude of physiological and pathological processes [27]. Exosomes are a subtype of EVs with a diameter of 30–150 nm and are made up of different molecular entities such as lipids, nucleic acids, and proteins and address it to specific recipient cells [28,29]. Recent research has revealed an expanded range of involvement of these lipid-bound nanovesicles in intracellular communication via transferring a wide variety of molecular cargos to recipient cells and coordination of biological events [23,30,31]. EVs are constantly released by a variety of cell types, including those grown in culture [32]. In addition, EVs are also present in biological fluids, including saliva, cerebrospinal fluid, breast milk, blood, urine, oviductal fluid in mice or fallopian tube in humans [29,30,31,33,34,35,36].

EVs are formed by inward budding of the endosomal membrane to construct multivesicular bodies (MVB) and released by cells following fusion of the MVB membrane with the plasma membrane [37]. However, few cell types can be induced to release EVs such as T lymphocytes and B lymphocytes via stimulation of cell receptors (TCR and BCR, respectively) during immune synapse. Once released into their surrounding environment, EVs can be taken up by the recipient cells via different mechanisms including clathrin-dependent endocytosis, proteoglycans [38] phagocytosis, macropinocytosis, [39] and fusogenic mechanisms facilitated by integrin receptors on EVs and the recipient cells [30]. Importantly, EVs can influence the signaling pathways of the target cells and affect their physiological or pathological processes [40]. It has become evident that EVs allow the exchange of complex information; thus, they are considered natural delivery systems involved in cell-to-cell communication and content exchange. Regarding EVs interaction with membrane receptors on target cells, innumerable proteins have been reported to be involved in this mechanism. EVs derived from the oviduct (oviductosomes, OVS) carry αvβ3 and α5β1 integrins on their surface and are captured to bind on sperm membranes. A decrease in OVS uptake by sperm cells was observed during the simultaneous inhibition of integrin/ligand interactions by antibodies, exogenous ligands, and their RGD recognition motif [30].

Another major milestone in field EVs was the discovery that EVs contain both mRNA and miRNA, which can be delivered to a recipient cell and influence the function of the recipient cell [32]. Fereshteh et al. [41] demonstrated that co-incubation of OVS with sperm can lead to OVS miRNAs (miR-34c-5p) transfer to a specific location on sperm at the centrosome to perform their functional activity.

EVs derived from the same parental cell are homogenous in their cargo and size. However, some cargos are partially common in EVs derived from various origins [42]. EVs contain numerous functional biomolecule cargos including proteins, mRNAs and miRNAs, making them capable of transmitting signals to target cells [30,41,43]. According to various EVs studies, they contain 9769 proteins, 3408 mRNAs, 2838 miRNAs, and 1116 lipids [34]. The surface of EVs is characterized by the presence of multi-sets of proteins involved in membrane adhesion and fusion (integrins, lactadherin, annexins), proteins associated with EVs biogenesis (Alix, TSG101, ESCRT complex), heat shock proteins (HSP70, HSP90), tetraspanins (CD63, CD81, CD82, CD9, TSPAN6, TSPAN8), and cell-type-specific proteins (MHC-I, MHC-II), etc. [42,44,45,46]. Moreover, OVS were shown to carry plasma membrane Ca^2+^-ATPase 4 (PMCA4) [36] as shown in Figure 1. The presence of anti-phagocytosis protein (“do not eat me” signal, CD47) on the EVs surface enables them to escape the phagocytosis by macrophages and reduce their clearance from circulation [47,48]. In addition, EVs carry all RNA species (miRNA, mRNA, transfer RNA, long non-coding RNA) [49,50]. Most lipids in EVs are known to be components of the plasma membrane, such as phosphatidylserine (PS), cholesterol (CHOL), sphingomyelin (SM) [44] as shown in Figure 1. Skotland et al. [51] reported the lipid composition of EVs isolated from cell cultures and bio-fluids contained PS and phosphatidylethanolamine (PE) on the inner leaflet of the EVs, while the outer leaflet had a high level of CHOL and SM. Furthermore, it is highly important to analyze the lipid profile in EVs to get insight into the lipid signature and to design a better carrier for drug delivery.

### 1.3. Difference between Natural EVs and Engineered EV-Mimetics in Therapeutics Delivery

The scientific community has progressively explored the utilization of EVs as ideal bio-carriers for gene and drug delivery, in particular against OC [52,53,54]. EVs have a pivotal involvement in intracellular communication; transporting their cargos to recipient cells, and overcoming clearance by macrophages [55]. Their clinical translation as a drug delivery system has been slow which may be as a result of: (i) batch-to-batch variation due to lack of standardized isolation and purification methods, (ii) potential viral contamination from donor cells, (iii) lack of scalability to meet the needs of a large population and (iv) time-consuming isolation processes [56,57]. Moreover, several attempts have been made to load tumor-derived EVs with drugs by applying different stress to cells, such as sonication or extrusion. The use of tumor-derived EVs in OC treatment has raised several safety concerns, such as their unknown molecular compositions and vesicles orientation; this is due to the extrusion process of EVs preparation which turns the membrane of the EVs inside-out thereby terminating their clinical application [51].

Although research utilizing EVs as a delivery vehicle for nucleic acid payloads is in its infancy phase, it can be beneficial to learn from the above-mentioned obstacles in the development of artificial mimetic EVs (Figure 2), which is also an emerging field for therapeutic delivery [58]. García-Manrique et al. [58] systematically classified artificial mimetic EVs as: (i) EVs-based semi-synthetic nanoparticles and (ii) EV-mimetic nanoparticles, based on types of modification and technology. The first classification, termed “EVs-based semi-synthetic nanoparticles”, is produced by the bioengineering of natural EVs before or after their isolation via multiple included surface and membrane modifications and could also include encapsulation with different cargos. The second classification, termed “EVs mimetic nanoparticles”, is artificial structures including those generated from cultured cells using top-down synthesis to increase the yields or lab-made using well-known molecules with only desirable function via bottom-up synthesis, which we hereafter refer to as “engineered synthetic EV-mimetic” [58]. The purpose of using artificial mimetic EVs as a drug delivery system is to achieve a simpler system with fewer molecular compositions for successful therapeutic application and commercial approval [58]. Owing to the similarity between both systems in terms of size (a range of 50 to 120 nm), shape (spherical lipid-bilayer nanoparticles) [58,59], and the incorporation of main functional components of natural EVs, including lipids, proteins, and therapeutic cargos [60]. Fully synthetic EVs or engineered synthetic EVs are an ideal approach to achieving simpler compositions than natural EVs for the successful therapeutic application and commercial approvals [58].

Engineered synthetic EVs can be produced by bottom-up synthesis, assembling individual molecules (lipids, proteins, and cargo) into complex structures, such as lipid bilayer structures resembling natural EV membranes functionalized with proteins for mimicking natural EV functions [60]. These engineered synthetic EVs may also contain specific proteins that allow efficient cell targeting and cellular uptake, which natural EVs possess [59,60]. Recently, Vázquez-Ríos et al. [61] designed liposomes based on specific molecular characteristics of EVs to enhance their targeting efficiency and immunogenicity. Sood et al. [62] also demonstrated that neutral liposomes produced by neutral lipid constructs, such as 1,2-Dioleoyl-sn-Glycerol-3-Phosphatidylcholine (DOPC) are desirable for nucleic acid delivery and this work led to a successful first-in-human Phase I clinical trial (identifier: NCT01591356). For many years, poly-ethylene glycol (PEG) and it conjugates have been the major ingredients for many liposomal formulations, its attractiveness stems from its superior biocompatibility, excellent solubility and high stability [63]. PEG is also used to improve the blood circulation time of liposomes and can be incorporated on the liposome surface via a cross-linking lipid [64]. For example, Figure 2 depicts an EV-mimetic with PEG polymer anchored to the liposome membrane (PEG-distearoylphosphatidylethanolamine (DSPE)). Thus, considering the above conditions, engineering synthetic EVs holds the promise of improving the delivery of bioactive molecules with therapeutic efficacy while achieving scalability and increasing bioavailability.

## 2. The Roles of Natural EVs in OC

EVs can be utilized in a wide range of applications for OC patients, including diagnostic and therapeutic tools, such as regenerative therapy, immunomodulation, drug delivery and antitumor therapy [52,53,54,65]. Despite the substantial contribution of EVs in cancer treatment to improve personalized medicine protocols in patients, there are a limited number of studies applying the benefits of EV research to OC therapy [66]. It is likely that secreted factors and EV cargo differ between various subtypes of OC. Therefore, it is essential to understand the nature of OC EVs to overcome the limitation of EV use in diagnostics, and treatment [67,68]. It is also critical to thoroughly characterize EVs isolated from OC cells to expatiate on the information reported by the International Society for Extracellular Vesicles (ISEV), in terms of morphology, size, presence of EVs biomarkers, etc. [69].

### 2.1. OC Progression

EVs are described as multi-signaling messengers involved in multiple physiological and pathological processes that can transfer their cargos, such as double-stranded DNA, mRNA, non-coding RNA (IncRNA and microRNA) and functional proteins [30,70]. For example, EVs can mediate cell-to-cell communication and intracellular signaling [21,71]. EVs play a role in stimulating cancer progression as their content, normal trafficking and functions are altered [72], resulting in the development of a cancer environment [73,74,75] leading to drug resistance.

EVs derived from immune cells, such as macrophages, dendritic cells, B-cells and T-cells, may mediate adaptive immune responses to pathogens and tumors [65]. They also have an essential function in cancer development, as tumor cell-derived EVs can transfer their oncogenic information to recipient cells allowing the progression, survival, metastasis and drug-resistant nature of tumors [76]. The remodeling of the extracellular matrix, the ability to promote angiogenesis, tumor vascularization, hypoxia-mediated inter-tumor communication and proliferation of cancer cells are other crucial roles of EVs in cancer progression [77,78]. EVs derived from Hela cells contain a high quantity of long non-coding RNAs-p21 (lncRNAs-p21) in comparison to the parental cell [79], which inhibits p53 (tumor suppressor gene) expression in recipient cells [80]. EVs derived from OC patients have a high concentration of miR-30a-5p and its knockdown showed inhibition of OC proliferation and migration [81]. OC EVs express low levels of miR-101, which is known to attenuate brain-derived neurotrophic factor (BDNF)/tyrosine kinase B (TrkB) signaling and thus inhibits OC invasion and proliferation [82]. On the other hand, epithelial OC EVs are enriched with miR-200b, which inhibits cell proliferation and enhances apoptosis in OC cell lines: SKOV3 and OVCAR3 [83]. The abundance of the let-7 and miR-200 family in EVs may be related to the invasiveness of OC [84]. EVs play a role in tumor angiogenesis as they have been shown to transfer metastasis-associated lung adenocarcinoma transcript 1 (MALAT1) from epithelial OC to human umbilical vein endothelial cells (HUVECs), resulting in promoting HUVECs angiogenesis [85]. Extracted EVs from OC cells are able to suppress T cell function through translocation of nuclear factor kappa B (NFκB), production of interferon (IFN)-γ, and expression of ganglioside on the EVs surface [86,87]. These extracted EVs also control the transcription of signal transducer and activator of transcription 3 (STAT3) and NFκB in lymphocytes [88] and they inhibit the action of natural killer cells (NKs) [89]. OC EVs transfer CD44 to mesothelial cells and increase miR-99a-5p expression in mesothelial cells, which stimulates secretions of matrix metallopeptidase 9 (MMP9), vitronectin and fibronectin, respectively, resulting in OC cell invasion [25,90].

Tumor-derived EVs have a role to play in the cancer progression by carrying immunosuppressive molecules such as growth factor beta 1 (TGF-β1), NK-cell ligands and programmed death-ligand 1 (PD-L1) [91]. Studies have shown that EVs derived from tumor-associated fibroblasts can enter OC cells promoting migration, invasion and epithelial–mesenchymal transition (EMT) through transformation of TGFβ1-mediated SMAD signaling pathways [92]; therefore, inhibiting the entry of these exosomes into OC cells may be a potential therapeutic approach.

### 2.2. OC Diagnostics

Tumor cell-derived EVs could be utilized as biomarkers for the diagnosis and prognosis of their originated cancerous cells, including OC [52]. Several reports have suggested that EVs are able to mediate tumorigenesis, metastasis (including OC peritoneal dissemination) and drug resistance in OC through the transferring of their contents, i.e., proteins, double-stranded DNAs, mRNAs, miRNAs and lipids [93]. These contents are considered biomarkers for detecting OC, OC stage and response to treatment. Many proteins have been identified from ovarian tumor-derived EVs, such as membrane proteins (Alix, TSG 101), tetraspanins (CD24, CD44, CD63, CD37, CD53, CD81), epithelial cell surface antigen (EpCAM), proliferation cell nuclear antigen (PCNA), tubulin beta-3 chain (TUBB3), epidermal growth factor receptor (EGFR), apolipoprotein E (APOE), claudin 3 (CLDN3), Claudin-4 (CLDN4), TrkB, fatty acid synthase (FASN), Erb-B2 receptor tyrosine kinase 2 (ERBB2), and cell adhesion molecule L1 (CD171), as well as enzymes (phosphate isomerase, peroxiredoxin, gelatinolytic enzymes, aldehyde reductase) [68,93,94,95,96,97]. Early detection of platinum-based therapy resistance can enhance the treatment outcomes for OC patients. Therefore, it has been reported that an increase in the expression of annexin A3 (ANXA3) protein may indicate platinum resistance in OC [95,98]. A study by Taylor and Gercel-Taylor [53] stated that miRNA profiling of the circulating ovarian tumor-derived EVs can be a potential marker for OC. Detected miRNA in EVs used as diagnostic markers for OC include miR-21 [83,99], miR-30a-5p [81], miR-373, -200a, -200b, -200c [100], reduced expression of miR-101 [82], circulating miRNAs (miR-200a-3p, miR-766-3p, miR-26a-5p, miR-374a-5p, miR-142-3p, let-7d-5p, miR-130b-3p and miR-328-3p) [101], miR-200b [83], and miR-1290 [102]. Additionally, detected lncRNA exosome such as MALAT1 is considered a diagnostic marker for OC [85]. Other biomolecules such as PS, glycans and glycoproteins have been reported to play a vital role in the internalization of EVs and signal recognition [103].

### 2.3. The Role of EVs in Delivery of OC Therapeutics

EVs not only play a vital role in cancer diagnosis and prognosis, but also present potential options for the treatment of OC. EVs can be utilized as novel therapeutic systems owing to their biocompatibility, bioactivity, high stability in the systemic circulation, high specificity to their target cells, resistance to different metabolic processes, ability to tolerate the immune system, and capability to penetrate impermeable biological barriers such as the blood–brain barrier (BBB) [43,104]. EVs could be loaded with different therapeutic agents including small molecules, proteins, peptides and nucleic acids. In 2005, several studies reported the first clinical trials utilizing EVs to treat non-small lung cancer and metastatic melanoma via subcutaneous/intradermal administration of antigen-loaded autologous dendritic cell EVs [105,106].

Modification of the EVs by engineering of its surface markers can be used to activate the immune system. Shi and colleagues managed to develop an exosome platform entitled synthetic multivalent antibodies retargeted exosome (SMART-Exo) to provide dual antibodies against CD3 (T-cells) and human epidermal growth factor receptor 2 (HER2) or EGFR in breast cancer [107].

Delivering therapeutic molecules to OC has challenges in terms of specific targeting, reduced immunogenicity reactions, and prolonged circulation time [59]. However, EVs have been demonstrated to be a promising carrier for small interference RNA (siRNA), which is a genetic therapeutic approach to down regulating gene expression with minimal side effects [108]. siRNA is easily degradable and needs a carrier vesicle for transportation to the targeted tissue and to enhance entering cells [109]. However, nanoparticles such as Poly (D/L-lactide-co-glycolide acid) (PLGA) polymer [109,110] and liposomal -based nanoparticles [111,112] have been successfully used to deliver siRNA to OC in vitro and in vivo. The siRNA molecule has been investigated as a candidate for potential gene therapy for OC, by targeting cell survival and drug-resistant genes such as multidrug resistance gene 1 (MDR1) and B-cell lymphoma 2 (Bcl-2) [113,114,115]. In addition, siRNA is used to target proliferation and angiogenesis pathways such as the vascular endothelial growth factor (VEGF) and ephrin type-A receptor 2 (EphA2) [116,117]. Furthermore, CD47 expressed on the EV membrane improved the circulation half-life of EVs by protecting them from phagocytosis [118]. In addition to siRNA, the permanent knockout of genes of interest could be achieved using the CRISPR system, where the targeting locus is recognized by the guide RNA, while the Cas protein in the CRISPR system causes a double-strand break with insertion/deletion (indels) effect [119,120]. To explore this approach, Kim and colleagues [121] induced an ovarian xenograft model using the SKOV3 cell line. Then, EVs of cancer origin were used to carry the CRISPR system targeting poly (ADP-ribose) polymerase-1 (PARP-1) gene. The mouse model was monitored for 20 days, results showed that mice treated with CRISPR EVs showed a significant reduction in tumor size compared to the control.

miRNAs are single-strand non-coding small RNA molecules that regulate gene expression [122]. In cancer cells, miRNA expression has a marked effect on tumor progression, invasion, and angiogenesis [123]. Regulating the miRNA signals in the cancer tissue with using anti-miRNA molecules or delivering mimic miRNA molecules to reverse the effect is a promising therapeutic approach [124]. For instance, Liu et al. [125] reported that restoring miRNA-506 in the OC can inhibit cancer proliferation through the obstruction of the CDK4/6-FOXM1 signal. In addition, targeting miRNA (miRNA-21) which is associated with cisplatin resistance, can help with regard to chemotherapy response and tumor suppression [126].

Chemotherapy is an effective regimen for numerous cancers. To avoid the unwanted targeting of healthy cells, reduce drug dosage, and prevent mutation initiation, the exosome has been explored as a potential carrier for chemotherapeutic drugs to cancer tissue [127,128]. In the case of OC, Tang et al. [129] demonstrated that chemotherapeutic drugs (cisplatin or cisplatin/paclitaxel) encapsulated by EVs were able to prolong the survival of a murine model with no detectable side effects. Furthermore, EVs can help to overcome poor solubility of cytotoxicity compounds such as Triptolide. In this context, Liu et al. [130] loaded the Triptolide drug by sonication and ultrafiltration centrifugation to EVs extracted from the SKOV3 cell line, a type of HGSOC. This formulation was tested in vivo where they showed a more than two-fold reduction in ovarian tumor size compared to the use of the Triptolide-free drug group.

Despite the great research efforts in the development of naturally occurring EVs for therapeutic delivery, there have been several obstacles to future applications and clinical trials, which include challenges in the production at large scale as well as efficient separation methods. In addition, the naturally occurring EVs have limited drug loading efficiency and reproducibility [131,132].

## 3. Current Clinical Uses of Natural EVs in OC

EVs have been around for decades, and several preclinical studies have explored the use of natural and synthetic EVs as novel delivery systems and diagnostic tools. However, in the field of the OC, only three clinical trial studies are registered at ClinicalTrials.gov (accessed on 31 May 2021) that utilize EVs. The first study (identifier: NCT03738319) evaluates the non-coding mRNA and miRNA content of the EVs from epithelial OC by analyzing them through next-generation sequencing as a diagnostic tool biomarker. In the second study (identifier: NCT02063464), blood samples were taken from OC patients, in which the monocytes with other non-cellular components in the blood, such as EVs, were evaluated for the ability of these components to kill tumor cells in a similar manner as monocytes of healthy people. The third study (identifier: NCT02662621) mainly focuses on evaluating the stress protein that is called HSP70, which is found in the exosome of blood and urine from patients suffering from cancer in which part of the participants have OC stage III and IV. This study aims to find a method for the early diagnosis of patients with solid tumor malignancies by evaluating the level of HSP70 in EVs. Despite the advantages of using natural EVs, the reported clinical trials mainly focus on using EVs as a diagnostic tool. Our finding yielded no studies in clinical trials utilizing natural EVs for OC therapeutics, owing to the formulation limitations of natural EVs lacking efficient and reproducible encapsulation of the therapeutic drugs [132].

## 4. Engineered EV-Mimetics in OC Therapy

Engineered EV-mimetics consist of minimal components for recapitulating the key features of natural EVs for persistent circulation. These engineered synthetic EVs are capable of loading and delivering therapeutics to target cells. The first clinically used liposome-based product using DOPC to treat neoplastic meningitis Depocyt^®^ was approved in 1999 [133]. Other formulations utilizing the neutral liposome have been made for treatments of hepatitis, influenza and for pain management as detailed by Bulbake et al. [134]. There have been several clinically approved liposomal formulations for OC chemotherapy such as Paclitaxel [135], and Doxil [136], which was approved for treatment of recurring OC in 1999. Doxil is also administered to patients with platinum resistance [137].

To date, there are no novel clinically approved drugs that utilize EV-mimetics for the delivery of OC therapeutics. A search for clinical trials targeting HGSOC at clinicaltrials.gov (accessed on 31 May 2021) [138] revealed nine results (three completed with no results, one terminated and five recruiting). All recruiting trials interestingly involve combination therapy with pegylated liposomal doxorubicin (PLD). There is therefore a need for the development of novel therapeutics utilizing the benefits of EV-mimetics for the treatment of HGSOC. Due to the advances in the field of EV-mimetics, there have been several promising studies and a recently concluded first-in-human phase 1 clinical trial (identifier: NCT01591356), which tested EphA2-targeting DOPC-encapsulated siRNA on patients with solid tumor malignancies [62]. Mice and mammalian safety studies for these synthetic EV-mimetics showed that the drug was tolerable at all doses [62].

Studies by Landen et al. [139] showed that the neutral liposome DOPC in conjugation with EphA2 targeting siRNA and paclitaxel significantly reduced tumor growth in the orthotropic mouse model of OC compared to using paclitaxel and siRNA alone. Mangala et al. [140] also demonstrated that platinum resistance ovarian carcinoma was significantly reduced by treatment using siRNA in conjugation with DOPC. The study by Chakravarty et al. [141] revealed that nuclear receptor co-regulator (PELP1)-siRNA incorporated into DOPC nano-liposomes significantly reduced tumor nodules, tumor growth, and ascites volume in OC cells expressing PELP1-shRNA (short-hairpin RNA). Wang et al. [142] successfully engineered a co-loaded paclitaxel and MDR1-siRNA biomimetic lipid/dextran hybrid nanocarrier as a drug and gene dual-delivery system against highly paclitaxel-resistant cancer cells. These nanocarriers, with a diameter of 100 nm, were able to promote the accumulation of the anticancer drug through knocking down the multi-drug resistance gene 1, MDR1, using MDR1-siRNA. Both in vitro and in vivo results against highly paclitaxel-resistant human OC cells and tumor-bearing BALB/c nude mice, respectively, indicated that the gene and chemo co-delivery by a biomimetic system has enhanced therapeutic efficiency against the highly paclitaxel-resistant cancer cells compared to the free paclitaxel [142]. Therefore, the use of EV-mimetics holds great promise as safe and effective delivery vehicles of therapeutics to solid tumor malignancies, especially HGSOC.

## 5. Tumor-Derived EVs vs. Engineered EV-Mimetics for Delivery of Therapeutic Agents

In recent years, tumor-derived EV-based semi-synthetic nanoparticles have been used as delivery vehicles for antitumor molecules such as siRNA, miRNA, and chemotherapeutics [143]. Tumor-derived EVs have the natural capability to target cancer tissue to stop the progression of cancer cells via the induction of the apoptosis cascade, the activation of the phagocytosis of tumor cells, and the inhibition of migration and invasion [144]. The novel properties of EVs in selective targeting of peritoneally disseminated OC cells indicates their therapeutic importance [145]. The study by Reza et al. [146] reported that the application of EVs isolated from adipose mesenchymal stem cells into A2780 human OC cells showed significant inhibition in the cell proliferation through the cell cycle blocking and mitochondria-mediated apoptosis signaling activation. Recently, Pisano et al. [147] evaluated doxorubicin-loaded immune derived exosome mimetics (IDEM) as a scalable delivery system against OC. These EV-mimetics demonstrated high drug entrapment efficiency, prolonged drug release profile and high OC cellular uptake. In addition, the in vitro cytotoxicity and apoptotic effect of doxorubicin-loaded IDEM were shown to be more efficient than the free anticancer drug [147]. Moreover, tumor-derived EVs have demonstrated an essential role in regulating the immune response to prevent the tumor progression and efficiently transport different cargos to the cell of origin. For instance, ascites-derived EVs carrying OC-specific antigens (e.g., HSP70, HSP90, MHCI, and Her2/Neu) are reported to induce the immune response via the activation of the NK cells and cytotoxic T cells in OC, therefore reducing the tumor progression [148]. However, the selectivity of tumor-derived EVs to the targeted tissue could be improved by performing surface modification or ligand conjugation, which leads to the more efficient delivery of cargo to the tumor tissue with minimal off-target effect.

The engineering of tumor-derived EVs via conjugation of tumor-targeted peptide (iRGD) using transfecting plasmid to form iRGD EVs was shown to target the NuTu-19 OC cells selectively via the membrane fusion mechanism [95]. The intravenous injection of doxorubicin-loaded into iRGD EVs in OC mice exhibited a significant reduction in the tumor volume when compared with the tumor treated with the doxorubicin or tumor-derived EVs alone [149,150]. The surface modification of tumor-derived EVs via the conjugation of staphylococcal enterotoxin B showed significant activation of apoptosis rate in SKOV3 cells, as well as proliferation reduction [151].

The engineering of EVs at the molecular level can direct the EVs into the immunotherapeutic pathway. The systemic administration of semi-synthetic EV-mimetics loaded with CRISPR/Cas9 targeting poly(ADP-ribose)polymerase 1 (PARP-1) demonstrably reduced the PARP-1 expression significantly in ovarian tumor tissue, thereby reducing the tumor progression [152].

Despite the therapeutic importance of tumor-derived EVs in the treatment of OC and targeting peritoneally disseminated OC cells, many challenges regarding the administration and the clinical application still lie ahead. One of the main concerns of tumor-derived EVs is the clearance from the circulation upon the recognition by macrophages and the immune system [153]. Moreover, the low tumor cell production of EVs and low loading efficiency of biomolecules are major limitations of using tumor-derived EVs that reduce their therapeutic efficiency [154]. Furthermore, the successful translation of tumor-derived EVs for clinical use faces the obstacles of scale-up production, difficulties in isolation and modification and targeted cell specificity [155,156].

Recently, engineered EV-mimetics have attracted scientific attention in the field of nanomedicine and drug delivery owing to their potential advantages and beneficial features as a delivery vehicle for biomolecules and therapeutic agents [157]. Engineering EV-mimetics offer distinct advantages in the treatment of OC over the tumor-derived EVs and the anti-ovarian cancer conventional therapy [158]. The manipulation of engineered EV-mimetics can improve the circulation and stability characteristics of the delivery vehicle following administration by protecting cargos from the opsonin proteins involved in phagocytosis and other immune components [132]. Engineered EV-mimetics have additional beneficial features such as surface modification capabilities, controlled composition, higher loading efficiency, lower immunogenicity, higher yield production, and sustainability [159,160].

Researchers have attempted to manipulate the potential features of surface decoration of nanoparticles in the engineering of EV-mimetics to improve cellular internalization through targeted cells. The conjugation of different transmembrane proteins and ligands can be implemented to reduce the interaction of nano-sized EVs with immune system components, thereby avoiding the subsequent clearance and eventual degradation by macrophages.

## 6. Engineering EV-Mimetics as Nano-Carriers for Cancer Therapeutics

Due to the vital role EVs play in the progression of tumor, metastasis [33,161,162], diagnosis, prognosis of cancer and response to treatment [163], it is advantageous to closely mimic these native EVs in designing a successful therapeutic delivery vehicle. This would not only improve the accumulation of these EVs in the tumor vasculature effect but also tumor uptake and consequently efficacy of the delivered chemotherapeutic or RNAi. We suggest that future formulations for delivery of treatment for OC considering these few design parameters to achieve optimal efficacy and reduce cytotoxicity.

### 6.1. Size and Surface Charge

It is a well-known fact that there is a direct correlation between structural characteristics (such as size and surface charge) of a nano delivery vehicle and its effects on the pharmacokinetic and pharmacodynamic properties in a tumor environment. The size of the nanocarrier affects the clearance by the renal system and by the reticuloendothelial system (RES), which includes the macrophages of the spleen and liver [157]. Particles between 30 and 100 nm are small enough to penetrate the tumor vasculature and accumulate at the tumor site [157,164]. The surface charge of nanoparticles affects their interaction in biological environments. Cationic liposomes such as dioleyltrimethylammoniumpropane (DOTAP) are also more toxic than neutral ones and have low fusogenicity [165]. To overcome this limitation, cationic liposomes are often conjugated with PEG, a hydrophilic polymer which is known to prolong circulation time. However, on reaching the cell, PEG has to be removed to permit endosomal escape and cargo delivery. Therefore, an ideal nanocarrier should be either neutral or negatively charged [166,167], as these have shown good bioavailability and pharmacokinetic profile, but with the drawback of interacting poorly with negatively charged siRNA [168].

### 6.2. Receptor-Mediated Targeting

Engineered EV-mimetics can promote cellular uptake across targeted cells, increase the vesicle stability, and improve the circulation time in the bloodstream [169]. Recently, engineered EVs have been shown to shorten the time required to reach the therapeutic concentration in targeted tissues and significantly enhance their accumulation in target tumor tissues and enhanced their therapeutic efficacy. For example, exosomes derived from normal fibroblast-like mesenchymal cells were engineered to carry siRNA specific to oncogenic KRAS-mutant pancreatic cancer cells and named “iExosomes”. These exosomes carry CD47 on their surface, which is known to interact with signal regulatory protein α (SRP α) on macrophages to enable “do not eat me” signaling and were found to evade phagocytosis by circulating monocytes and macrophages. They also increased their ability to deliver RNAi to pancreatic cancer cells and suppress tumor growth [118]. It has been shown that conjugation of this minimal peptide to synthetic nano-beads impeded macrophage-mediated phagocytosis of nano-beads. Therefore, the availability of minimal peptides from both human and mouse CD47 will enable potential translation of mouse studies into humans (Figure 3) [170,171]. Moreover, homing ligands, such as tetraspanins (CD9 and CD63), integrins, and connexins, conjugated on the EV surface can enable EV targeting and have reportedly enhanced the intracellular internalization of EVs loaded with biomolecules through the fusion through the cell membrane [169,172]. They have also been used for targeted delivery of drugs and RNA therapeutics [149,173,174,175,176]. These proteins exhibit a crucial characteristic in facilitating the intracellular delivery of biomolecules across several conditions (i.e., OC). The conjugation of cell-penetrating peptide (CPP) and poly(ethylene glycol) (PEG) demonstrated to enhance the selectivity of EVs to targeted cells, improve the circulation time and increase colloidal stability [93,177]. Therefore, engineered EV-mimetics can successfully adapt lipid–peptide conjugation chemistry to incorporate desired peptides (with a broad range of densities) in a well-controlled manner, which cannot be achieved by genetic engineering.

To improve the therapeutic effect, engineered EV-mimetics should be functionalized with ligands that display specificity and selectivity for OC tumors. Target-based drug delivery of EVs has been of interest in the scientific community owing to the advantages of high bioavailability and increased therapeutic impact compared to passive targeting of the free drug, which relies on the enhanced permeability and retention (EPR) effect. OC epithelial tissues display several tumor-associated antigens (TAA) on their surface that differentiate them from healthy OC tissues. Hundreds of TAAs have been identified over the years, which include Mucin-16 (MUC1–16) (CA125), Human Epididymis Protein 4 (*HE4* gene), Alpha-Fetoprotein (AFP), Osteopontin (OPN), tumor protein 53 (p53), Kallikrein-6 (KLK6), Mesothelin (MTLN), plasminogen activator, tissue type (PLAT or tPA), homeobox protein Hox-A7 (HOAX-7) and Interleukin-8 (IL-8) [178,179,180,181,182,183]. These antigens show great promise as prospective targets for OC early diagnosis as well as therapy utilizing EVs conjugated to tumor-associated autoantibodies (AAbs). Two AAbs of note that show over 95% target specificity to TAAs are HOAX-7 AAbs for moderate differentiating OC and IL8-AAbs for detection of stage I to stage II OC [180].

Tumor-associated proteins are another promising target for chemotherapy or RNA therapy. As mentioned earlier, several proteins are overexpressed in EVs derived from OC and are used as a potential diagnostic tool. One of such proteins is the folate receptor (FR) which is a glycosyly phosphatidinositol (GPI)-anchored membrane protein over-expressed in 90% of OC and thus is an attractive target for drug delivery. Yang et al. [184] demonstrated successful tumor in vitro and in vivo targeting using folate ligand, which is the ligand for FR, and the formulation which included the co-delivery of DOX and Bmi1 siRNA by folate-DOX/Bmi1 siRNA liposome effectively prevented tumor growth (Table 1).

Hong et al. [185] described a novel approach to nanoparticle targeting of OC (Table 1). Their strategy was to equip PEG-PEI particles with a short peptide mimicking the sequence of the follicle-stimulating hormone (FSH) and loaded with siRNA against the growth-regulated oncogen α (gro-α), (sigro). PEG-PEI-FSH-sigro particles delivered to Hey cells, which express both FSHR and gro-α, reduced the target mRNA level down to 47.3% of that of the control. Notably, when PEG-PEI-FSH-sigro was delivered to SKOV3 cells, which lack FSH receptors, a reduction in target mRNA levels by 79% was observed, indicating the effectiveness of the targeting strategy.

A protein commonly expressed on tumor cells is EGFR and is a common target in other cancers such as colorectal and breast cancer. Epidermal growth factor (EGF) is a stable protein, with well-defined conjugation sites. Wang et al. [186] were successfully able to construct a liposome with sodium alginate (SA) and Cisplatin (CDDP), an antineoplastic drug for OC loaded into its hydrophilic core, and incorporated EGF to produce a CDDP-SA-EGF-basic liposomal complex (CS-EGF-Lip) (Table 1). The authors were able to successfully prove that this complex effectively reduced tumor growth in SKOV3 in vivo and in mice models with a reduction in nephrotoxicity in comparison to CS-PEG-Lip alone [186], thus effectively demonstrating the advantage of membrane targeting.

An additional strategy to target endothelial OC cells was described by Kim et al. [187]. The authors targeted the CD44 receptor and tumor endothelial marker 7 (TEM7), also known as plexin domain-containing 1 (PLXDC1). CD44 [188,189] and PLXDC1 [190] are overexpressed in OC-associated endothelial cells. PLXDC1 is known to be involved in the promotion of cell migration and invasion of tumor endothelial cells [191,192]. Kim and colleagues generated chitosan (CH) particles coated with hyaluronic acid (HA) (Table 1), which is a ligand that can bind specifically to CD44 receptor [193,194]. Additionally, these NPs loaded with siRNA against PLXDC1. This complex enables endocytosis intracellular trafficking [195,196]. The “spongy effect” enables CH-NPs to efficiently release siRNA from endosomes/lysosomes to the cytoplasm [197]. They delivered their particles in vitro to two models of endothelial cells (HUVEC and MOEC) and showed significant inhibition of cell migration and invasion due to PLXDC1 silencing. The authors suggest this may be useful for anti-angiogenesis tumor therapy.

Byeon et al. [110] used a two-in-one combination strategy of co-delivering chemotherapeutics and siRNA using CD44 as the epithelial OC target. They encapsulated HA-PLGA-NP with paclitaxel (PTX) and siRNA directed against focal adhesion kinase (FAK) (Table 1), which plays a vital role in cell migration, invasion, and survival [198] and also is overexpressed in OC [199]. Significant inhibition of tumor growth was observed compared to treatment with PTX alone. These studies further confirm the important role of selective targeting delivery systems to overcome chemoresistance in OC therapeutics and highlighting the need for applying these strategies to improve treatment outcomes.

Mucins (MUC1, MUC3, MUC4, and MUC16) are heavily glycosylated tethered glycoproteins characterized by the presence of transmembrane domains and diverse signaling mechanisms [200]. They are overexpressed in OC tumors and have been well studied, especially MUC1–16, where its *N*-glycosylation sites were preserved in all of the HGSOC as listed in Cancer Genome Atlas (TCGA) [201]. MUC1–16 (CA125) is used as a biomarker for OC due to its overexpression which leads to the release of MUC1–16/CA125 in serum [182,202,203,204]. Notably, Rao et al. [203] demonstrated that the site-specific *N*-glycosylation of the MUC1–16 ectodomain is recognized by Galectin-3 antigen and induces the oncogenic transformation. Therefore, they have developed monoclonal antibodies (synthetic MUC16 glycopeptides) against MUC16 to block Galectin-3-mediated MUC16 interactions with cell surface signaling, and later reduce OC invasion and growth [203]. Thus, MUC16 could be a promising candidate for the treatment of HGSOC. Together, these studies provided insights into how molecular properties of nanoparticles may mediate particle-cell communication.

### 6.3. Intracellular Trafficking of EV-Mimetics

The trafficking pathway exhibited by liposome-based EVs is a complex process governed by the interactions between the physiochemical properties of the EV, their endocytosis mechanism and target cells [205,206,207,208]. Most research has focused on nanocarrier uptake mechanisms by cells and lysosomal escape [209,210,211]. However, the intracellular traffic of engineered EVs is crucial in the development of effective chemotherapeutic delivery strategies; Arta and colleagues demonstrated that there is no direct correlation between liposome uptake by a cell and drug efficacy, but rather it is liposomal intracellular trafficking, particularly liposome distribution into recycling endosomes and lysosomes that influences in vitro efficacy [212]. Their experiments elucidated the effect of common targeting strategies on the efficacy of the liposomal delivery of DOX to HRECs cells by comparing three common methods of cell targeting: monoclonal antibody (TfR-NLs), functionalized liposomes targeting the endothelial protein C receptor (EPCR-NL) and cationic small molecules (CAT-NL). Results show that TfR-NL was significantly distributed in early endosomes as well as recycling endosomes. CAT-NL liposomes were significantly located in lysosomes compared to other trafficking organelles after 4 h of incubation in HRECs. The results also showed CAT-NL had significantly higher release of DOX in the HRECs. A similar observation that nanocarriers distribute into lysosomes was made by Rejman et al. [213].

It is also well known that liposomes enter target tissues via direct fusion, such as with cationic or neutral fusogenic liposomes [214,215,216,217] or by endocytosis [218,219] (Figure 3). However, the fusion pathway of cationic liposome is inhibited upon addition of DNA, due to lipid-DNA electrostatic interactions. Thus, the only pathway of entry of cationic lipid/DNA complex is via endocytosis [220].

The best studied uptake pathway of liposome-based EVs is via clathrin-mediated endocytosis, whereby EVs bind to specific receptors in clathrin-coated pits [219,221,222]. Upon internalization, liposome-based EVs fuse with early endosomes, however the best studied pathway is via the clathrin-mediated vesicles. Although it is not well understood how the different components sort into their respective targets, the most acceptable model is the pathway from the early endosome, to late endosome and finally to lysosomes [223] (Figure 3). Early endosomes are destined for lysosomal degradation [224]. However, evidence suggests that post uptake, EVs are able to escape lysosomal degradation and exert functional effects via the delivery of their cargo, but the mechanism is still unknown [220]. One possibility would be after the internalization of cationic lipid/DNA complex into the cell by endocytosis; fusion may occur between the liposomal membrane and endosomal membrane, leading to destabilization of the endosomal membrane, thereby allowing the release of DNA into the cytoplasm [225].

Studies to elucidate the post uptake EV trafficking routes have proven difficult as commonly used lipophilic dyes for EV labeling, such as PKH26, form particles that are indistinguishable from labeled EVs and co-localize with them in subcellular compartments [226]. Additionally, methods of tracking of EVs by using fluorescent labeled tetraspanin proteins [227] are ineffective because the EVs remain associated with the membrane after the delivery of the cargo [228].

Un and colleagues [222] demonstrated that liposomal components might be transported via different intracellular trafficking process after clathrin-mediated endocytosis. The intracellular localization of DOPC derived from liposomes was to the endoplasmic reticulum (ER) and Golgi apparatus, which was partly controlled by oxysterol-binding protein-related protein 1 (ORP1). Whereas the intracellular localization of Chol derived from liposomes was to the ER, which was partly controlled by Niemann–Pick C1 protein (NPC1). The results indicate that phosphatidylcholine-based liposomes are degraded somewhere between the endosomes and the ER/Golgi apparatus. Therefore, when designing EV mimetics, scientists must be mindful that EV lipid components affect intercellular trafficking and efficacy.

### 6.4. Delivery of EV-Mimetics to Mouse OC Models

Mice models are used for in vivo validation of OC therapeutics. There are three common pathways to investigate OC growth, metastasis and treatment using mice: the xenograft model, the synergistic model, and the genetically engineered mouse model (GEMM). The xenograft model has been extensively used since the 1980s and involves injecting OC tumor cells of different genetic backgrounds into immune compromised animals such as Nude (Foxn1, Nu/Nu), SCID, NOD/SCID, or NOD-scid IL2Rcnull (NSG), which enables the cells to engraft without being eliminated by the immune system. The majority of OC xenografts involve implantation of ovarian cancer cell lines at three different locations subcutaneous (SC), intraperitoneally (IP), or intrabursally (IB, orthotopic) [230].

The IP and IB models are better for studying ovarian metastatic dissemination into the peritoneal cavity, however IP models are more frequently used. IB has the advantage of accurately reproducing the initial steps in OC metastasis. Early events in cancer metastasis cannot be fully recapitulated when IP xenografts are used because these xenografts are generated by injecting cell suspensions directly in the peritoneal cavity. In comparison, orthotopic OC xenografts mimic the tumor initiation from anatomically relevant loci. Existing approaches for orthotopic mouse OC models include, but are not limited to, injecting cancer cell lines under the ovarian bursa and implantation of human tumor tissues in the vicinity of the ovary [231].

The SC model on the other hand is more suitable for imaging purposes. Studies show that mice injected with SKOV3 cells IP metastasize to the ovary, peritoneal wall, diaphragm, and form ascites fluid similar to human disease [232]. However, the major limitation of IP and SC OC models is the location of tumor formation and their resemblance to the human disease. The disadvantage of all xenograft animal models is the inability to study the interactions of the drugs on the immune system due to the immune compromised state of the mice [233]. Goldberg et al. [234] used lipid-like materials known as lipidoids to successfully deliver siRNA IP to athymic nude BALB/c mice. They also demonstrated that it is possible to use immunocompetent mice to examine the effect of inhibition of specific genes on tumorigenesis by using FVB/NJ mice in which tumor growth was possible.

Biodistribution of EVs are critically influenced by their routes of administration, dose and cellular origin [235]. Takahashi et al. [236] showed that luciferase-loaded EVs with were rapidly cleared from blood circulation with a half-life of approximately 2 min after (intravenous) systemic administration in vivo. It was shown that the presence of a “do not eat me” signal, CD47, on EVs is responsible for reducing their clearance by macrophages via interaction with SIRP α on macrophages, whereas its absence led to faster macrophage-mediated clearance of EVs isolated from CD47-knockout cells, which is similar to that of synthetic liposomes (Figure 4) [118].

## 7. Conclusions, Future Outcomes and Challenges

OC affects millions of women globally and is referred to as a silent killer of women because it is often detected after the cancer has advanced and most often is in stage III. HGSOC is known to be disseminated throughout the abdomen and this a critical clinical issue as many valuable organs, including the breast, stomach, pancreas, gallbladder, small intestine, colon, and urinary bladder are vulnerable to adhesion and tumor invasion [237]. The past decade has seen advancement in nanotechnology, and its application in OC therapy [238,239]. There has been an increase in EV-based therapeutics for other malignant myopathies with several native EVs approved and some in clinical trials for breast cancer, colorectal and brain cancer. Their surfaces have also been engineered to overcome challenges associated with the use of native EVs as therapeutic carriers, such as purification, isolation, scale-up, yield, loading efficiency, long-term effects [129,130] and to improve therapeutic delivery for a variety of cancers. Despite the advances in target-based delivery, to date, barely any improvement has been seen in the survival of OC patients in the past 30 years [240,241]. Therefore, there is a need for target-based nano drug delivery formulated with specific ligands that target HGSOC tumors, especially in the peritoneal cavity rather than depending on passive mechanisms [204,242,243,244,245,246,247,248]. There is a growing interest in using EV mimetics, which are liposome-based nano-vesicle complexes that mimic the naturally occurring EVs in vivo. One ongoing challenge with engineering EV-mimetics is the discovery of surface proteins of natural EVs that are responsible for targeting and uptake by cancer cells. To this end, formulation of lipid–peptide conjugates that closely mimic the surface of natural EVs may represent a viable solution to this challenge. Studies also suggest that target delivery is not enough and EV-mimetics would need to be coupled with methods to sensitize aggressive or recurrent tumors to platinum chemotherapy in order to improve patient outcomes and survival rate after late-stage diagnosis [240]. To tackle this problem, EV-mimetics can be formulated for co-delivery of chemotherapeutics as well as RNAi against genes associated with platinum-based chemotherapy resistance. For instance, DOPC-derived liposomes have demonstrated superior preclinical performance in delivering siRNA in orthotropic ovarian cancer models [139,140,141].

With the success of EVs therapeutics in the treatment of other cancers, there is increased expectation that use of an engineered EV-mimetic for the delivery of novel chemotherapeutics, RNAi or immunotherapy or a combination of both options for the treatment of HGSOC would yield promising results and put an end to this silent killer. In the near future, we expect to see engineered EV-mimetics integrated with improved capabilities from advances in nanotechnology and bioengineering to produce a more efficient drug delivery system that will have improved bio-compatible membrane structures with low cytotoxicity, highly controllable and pharmaceutically acceptable for commercialization. These advances would lead to researchers adapting engineered EV-mimetics for treatment of other pathologies and abandon the unpredictable and difficult protocols associated with production of natural EV-based therapeutics.

## Figures and Tables

**Figure 1 cancers-13-03075-f001:**
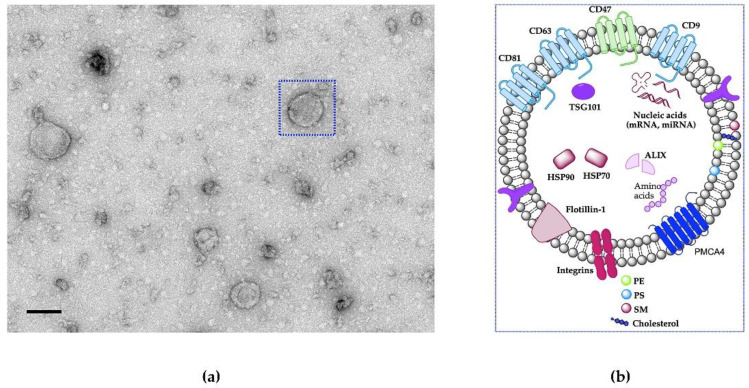
Natural EVs morphology. (**a**) TEM image of natural EVs (oviductosome, OVS) isolated from mouse oviductal fluid of ~100 nm in size. Scale bar represents 100 nm. (**b**) Schematic showing the structural components and cargo of extracellular vesicles which packed with a variety of cellular components, including various transmembrane proteins (tetraspanins, PMCA4), heat shock proteins, adhesion proteins, nucleic acids (mRNAs and miRNAs), and lipids (PE, PS, SM, and cholesterol). EVs also protect encapsulated cargo from clearance by macrophages via CD47 (“do not eat me” signal).

**Figure 2 cancers-13-03075-f002:**
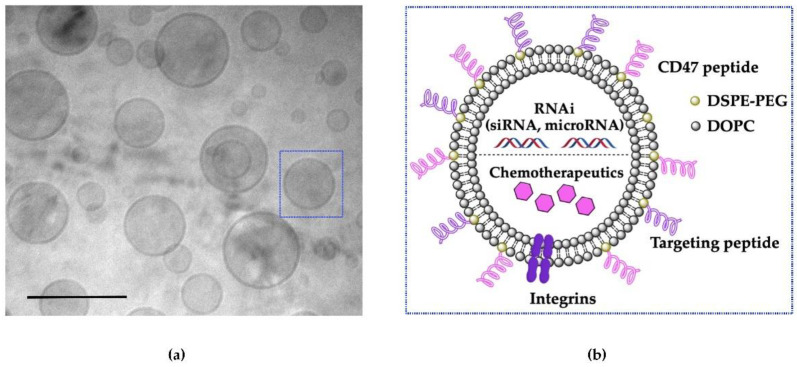
Engineered synthetic EV-mimetic morphology. (**a**) Cryo-TEM image of engineered synthetic EV-mimetic inspired by natural EVs lipid composition for therapeutic delivery of small molecules, proteins, drugs. Scale bar 200 nm. (**b**) Schematic illustration of surface functionalization of synthetic EV-mimetics by insertion of a conjugated lipid DSPE-PEG-CD47 and encapsulation siRNAs and chemotherapeutics for enhanced therapeutic effects.

**Figure 3 cancers-13-03075-f003:**
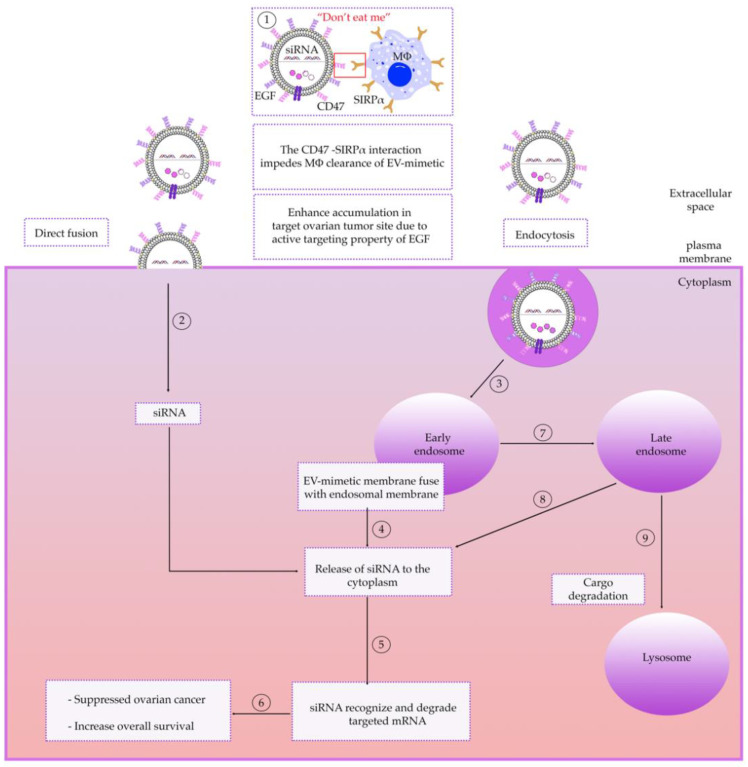
Schematic of the intracellular delivery of EV-mimetic with RNAi cargo to OC. Increased targeting of therapeutic EV-mimetic to OC cells and extension of their half-life of EV-mimetic in extracellular space can be achieved by coating EV-mimetics with molecules that enable specific targeting of tumor cells (such as EGF) or with molecules that enable escape of phagocytosis by macrophages (synthetic peptide CD47). Interaction of CD47 with SIRPα receptor on macrophages (MΦ) impedes the clearance of EV-mimetics and enhances accumulation in target tissue (1). After encountering the target cell, the EV-mimetic is typically bound to its surface via cell-surface receptors, integrins, etc. After establishing an interaction with the cell surface, EV-mimetics can be taken up by the OC cell via two mechanisms of internalization: firstly, involving direct fusion with the plasma membrane (2) or secondly via endocytosis; whereby EV-mimetics are taken up by early endosome (3). siRNA cargo is released into the cytoplasm (4). siRNA recognize and degrade targeted messenger RNAs (mRNAs) (5). This in turn leads to the suppression of OC and increases overall survival (6). Early endosomes will gradually transform into late endosomes (7) and possible release of siRNA to the cytoplasm (8). Further down the endocytic pathway, endosomes fuse with lysosomes, consequently cargo that has not been released to the cytoplasm will be degraded (9). Adapted from O’Brien et al. [229].

**Figure 4 cancers-13-03075-f004:**
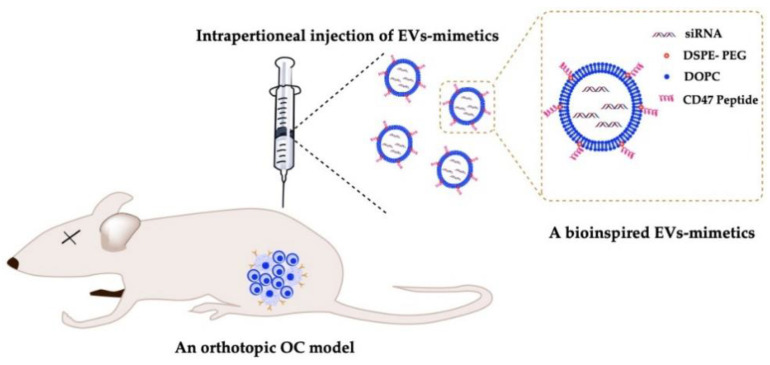
Delivery of EV-mimetics with RNAi cargo to mouse OC models. The CD47 peptides will help improve the pharmacokinetics by preventing macrophage-mediated phagocytosis, while the encapsulated siRNA induces gene silencing in ovarian tumors.

**Table 1 cancers-13-03075-t001:** Select target delivery systems against OC.

Delivery System	PM Target	siRNA Target	OC Therapeutic	Reference
PEG-PEI-FSH	FSHR	Gro-α	-	[185]
CS-EGF-Lip	EGFR		CDDP	[186]
HA-PLGA	CD44	FAK	Paclitaxel	[110]
HA-CHITOSAN	CD44	PLXDC1	-	[187]
DOTAP/Chol/mPEG-DSPE/FA-PEG-Chol	FR	Bmi1	Doxorubicin	[184]

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
