# Peer review of "Engineered EV-Mimetic Nanoparticles as Therapeutic Delivery Vehicles for High-Grade Serous Ovarian Cancer"

_cancers, 2021, doi:10.3390/cancers13123075_

Round 1

Reviewer 1 Report

Corrections and suggestions appropriately annotated, manuscript now suitable for publication

Reviewer 2 Report

The revised manuscript “Engineered EVs Mimetic Nanoparticles as Therapeutic Delivery Vehicles for High-grade Serous Ovarian Cancer” have adequately addressed my previous concerns and the paper is now acceptable for publication.

This manuscript is a resubmission of an earlier submission. The following is a list of the peer review reports and author responses from that submission.

Round 1

Reviewer 1 Report

Al-Dossary et al have presented a well written review on the potential use of extracellular vesicles (EVs) in targeted ovarian cancer therapeutics. I think the manuscript warrants publication with a few minor corrections. The main issue with this is that there is seemingly a paucity of information on the mechanistics on EV trafficking to the target cells. The authors point out mechanisms of uptake via integrins, MMPs etc, but how the EVs migrate to these cells needs further clarification. Furthermore, the manuscript could really benefit from a cartoon depicting the mechanisms behind EV transportation as well as the processes which occur on contacting the target cells. Other than this it was easy to read and certainly would be a suitable addition to the emerging literature in this field.

Minor comments

Line 137 there is a typo where overcom should be overcome

Line 146 neuclic should read nucleic

Line 451, after reference 165, there should be a comma as the sentence in its current form does not make sense

Line 528 Which the success should read with the success

Reviewer 2 Report

The review by Al-Dossary et al, describes in general the role of extracellular vesicles as delivery systems for cancer therapy.

The review does not add any new observations/conclusions based on the literature in comparison to the already existing reviews regarding extracellular vesicles as delivery systems for cancer therapy.

The review is titled as 'Engineered EVs Mimetic Nanoparticles as Therapeutic Delivery Vehicles for High-grade Serous Ovarian Cancer', however it is not focused on Ovarian cancer. Authors should survey the literature for use of Engineered EVs Mimetic Nanoparticles in Ovarian cancer and submit a more concise and focused version of this review to be suitable for publication.

Reviewer 3 Report

The manuscript “Engineered EVs Mimetic Nanoparticles as Therapeutic Delivery Vehicles for High-grade Serous Ovarian Cancer” by Amal A. Al-Dossary and co-authors demonstrated that natural occurring extracellular vesicles (EVs) have been shown to play a pivotal role in progression of OC and are now being harnessed as a delivery vehicle for cancer chemotherapeutics. However, there are limitations to their clinical application due to current challenges in their preparation techniques. Intriguingly, there is a recent drive towards the use of engineered synthetic EVs for the delivery of chemotherapeutics and RNA interference therapy (RNAi), as they show promise of overcoming the obstacles in the treatment of OC patients. This review discusses the therapeutic application of EVs in OC and elucidates the potential use of engineered EVs mimetic nanoparticles as a delivery vehicle for RNAi therapy and other chemotherapeutics, which would potentially improve clinical outcomes of OC patients. However, there are several important aspects which must be taken into account before the work can be reconsidered for publication.

Comment

1. The orthotopic (intrabursal) or intraperitoneal animal model is common research model for ovarian cancer. Please add into manuscript. How to apply EV in these animal models?
